# In Vitro Erythropoiesis at Different pO_2_ Induces Adaptations That Are Independent of Prior Systemic Exposure to Hypoxia

**DOI:** 10.3390/cells11071082

**Published:** 2022-03-23

**Authors:** Greta Simionato, Antonia Rabe, Joan Sebastián Gallego-Murillo, Carmen van der Zwaan, Arie Johan Hoogendijk, Maartje van den Biggelaar, Giampaolo Minetti, Anna Bogdanova, Heimo Mairbäurl, Christian Wagner, Lars Kaestner, Emile van den Akker

**Affiliations:** 1Department of Experimental Physics, University Campus, Building E2.6, Saarland University, 66123 Saarbrücken, Germany; antonia-rabe@gmx.de (A.R.); christian.wagner@uni-saarland.de (C.W.); lars_kaestner@me.com (L.K.); 2Department of Experimental Surgery, Campus University Hospital, Building 65, Saarland University, 66421 Homburg, Germany; 3Sanquin Research, Landsteiner Laboratory, Department of Hematopoiesis, Amsterdam UMC, University of Amsterdam, 1066 CX Amsterdam, The Netherlands; jsgallegom@gmail.com; 4Department of Biotechnology, Faculty of Applied Sciences, Delft University of Technology, 2629 HZ Delft, The Netherlands; 5Sanquin Research, Landsteiner Laboratory, Department of Molecular Hematology, Amsterdam UMC, University of Amsterdam, 1066 CX Amsterdam, The Netherlands; secretariaaty4@sanquin.nl (C.v.d.Z.); a.hoogendijk@sanquin.nl (A.J.H.); m.vandenbiggelaar@sanquin.nl (M.v.d.B.); 6Department of Biology and Biotechnology “L. Spallanzani”, Laboratories of Biochemistry, University of Pavia, I-27100 Pavia, Italy; minetti@unipv.it; 7Red Blood Cell Research Group, Institute of Veterinary Physiology, University of Zurich, CH-8057 Zurich, Switzerland; annab@access.uzh.ch; 8University Hospital Heidelberg, Medical Clinic VII, Sports Medicine, 69120 Heidelberg, Germany; heimo.mairbaeurl@med.uni-heidelberg.de; 9Translational Lung Research Centre Heidelberg (TLRC), Part of the German Centre for Lung Research (DZL), 69120 Heidelberg, Germany; 10Translational Pneumology, University Hospital Heidelberg, 69120 Heidelberg, Germany; 11Physics and Materials Science Research Unit, University of Luxembourg, L-1511 Luxembourg City, Luxembourg; 12Theoretical Medicine and Biosciences, Campus University Hospital, Building 61.4, Saarland University, 66421 Homburg, Germany

**Keywords:** in vitro erythropoiesis, hypoxia, high altitude, neocytolysis, pO_2_

## Abstract

Hypoxia is associated with increased erythropoietin (EPO) release to drive erythropoiesis. At high altitude, EPO levels first increase and then decrease, although erythropoiesis remains elevated at a stable level. The roles of hypoxia and related EPO adjustments are not fully understood, which has contributed to the formulation of the theory of neocytolysis. We aimed to evaluate the role of oxygen exclusively on erythropoiesis, comparing in vitro erythroid differentiation performed at atmospheric oxygen, a lower oxygen concentration (three percent oxygen) and with cultures of erythroid precursors isolated from peripheral blood after a 19-day sojourn at high altitude (3450 m). Results highlight an accelerated erythroid maturation at low oxygen and more concave morphology of reticulocytes. No differences in deformability were observed in the formed reticulocytes in the tested conditions. Moreover, hematopoietic stem and progenitor cells isolated from blood affected by hypoxia at high altitude did not result in different erythroid development, suggesting no retention of a high-altitude signature but rather an immediate adaptation to oxygen concentration. This adaptation was observed during in vitro erythropoiesis at three percent oxygen by a significantly increased glycolytic metabolic profile. These hypoxia-induced effects on in vitro erythropoiesis fail to provide an intrinsic explanation of the concept of neocytolysis.

## 1. Introduction

Erythropoiesis in the bone marrow is regulated by peripheral blood oxygen levels. Atmospheric oxygen pressure at sea level is 21 kPa (21%), which is reduced to 13% in the lungs and 3% in the extravascular space of the bone marrow, corresponding on average to a dissolved O_2_ concentration (dO_2_) of 0.6 mg/L [1]. Hypoxia induced by the reduced air pressure as it occurs at high altitude causes an increase in the rate of erythropoiesis, boosting the number of red blood cells (RBC) to ensure adequate oxygen delivery to the tissues. Such stimulation occurs through erythropoietin (EPO), which promotes proerythroblast proliferation and maturation. Transcriptional activity of the EPO gene is regulated by the oxygen-dependent degradation of the transcriptional factor HIF-2alpha [2]. High-altitude studies have shown initial increases in plasma EPO levels that subsequently decrease again but remain above sea level concentrations, allowing continuously elevated erythropoiesis [3]. Return to sea level results in a decrease in EPO and a drop in RBC mass up to 10% within 10 days [4]. A similar effect occurs in astronauts, who experience so-called “space anemia” during spaceflights. In this case microgravity causes a reduction in both plasma volume and 11% RBC mass within 10 days in order to compensate for the reduced peripheral vascular space (acute plethora) [5]. The adaptation of the hematocrit in both these conditions was previously judged to be too fast to be explained exclusively by the effect of EPO in reducing erythropoiesis rate, leading to the “neocytolysis” hypothesis. Neocytolysis is the selected removal of only young RBC (neocytes, formed at high altitude) upon return to sea level or upon ascent to space [6]. When occurring upon return from high altitude, some proposed mechanisms are putatively due to different characteristics of RBC derived from hypoxia-induced erythropoiesis. Parameters supporting the neocytolysis hypothesis are the detection of hemolytic markers (bilirubin and urobilinogen) [7], decreased expression of functional markers CD47, CD55 and CD59 associated with RBC clearance caused by EPO decrease [8,9], unchanged iron homeostasis [6] and a decreased reticulocyte count [7]. However, a recent study showed hemolysis to be the primary cause of space anemia, involving all RBC independent of their age, thus not supporting the neocytolysis hypothesis [10]. Moreover, markers associated with hemolysis were not observed in many studies at high altitude [3,5]. Indeed, we and others have performed a high-altitude study where no evidence for neocytolysis was observed [11]. Ever since, this hypothesis has become even more controversial [12,13,14,15,16,17], and the mechanisms acting on RBC lifespan in response to hypoxia are not fully understood, while the effects of hypoxia on erythropoiesis itself have been at best ill-described. Understanding the influence of hypoxia on erythroid development is important to characterize the homeostasis of RBC. The lack of an exact definition of “neocytes” (RBC up to 7 days old and including or excluding reticulocytes depending on the study), together with the experimental difficulties faced at high altitude or in space have led to incomplete and inconsistent data from various studies that, in addition, focused on RBC and not directly on erythropoiesis, which is the sole determinant of RBC intrinsic characteristics.

Here we studied in vitro erythropoiesis at low and high oxygen (O_2_) pressures and from peripheral blood mononuclear cells (PBMC) isolated before and at high altitude (3450 m) to better characterize the effects of O_2_ saturation on erythroid development. Besides assessing the functionality of cultured RBC (e.g., deformability, morphology and hemoglobinization), we performed unbiased proteomics on the resulting cultured RBC. Minimal differences between in vitro erythropoiesis of “high altitude” samples and control samples were found. As expected, metabolic adaptation occurred upon in vitro erythropoiesis at three percent O_2_, adjusting to a more glycolytic profile compared to control and altitude in vitro erythropoiesis at 20% O_2_. Overall, cultured RBC from three percent O_2_ were highly comparable to 20% O_2_ control cultured RBC. The study suggests that high altitude hematopoietic stem and progenitor cells do not retain a hypoxic footprint, but that erythropoiesis at low O_2_ (three percent) results in metabolic adaptation.

## 2. Materials and Methods

### 2.1. Design of the Study

Twelve healthy male volunteers in their twenties, living in Heidelberg (110 m above sea level), were recruited for the study after written informed consent. The study was approved by the ethics committees of the University of Heidelberg, Germany, (S-066/2018) and of the University of Berne, Switzerland (2018-01766) and was performed according to the Declaration of Helsinki. The volunteers were exposed to a 3-week stay at the High Altitude Research Station at the Jungfraujoch (“JJ”, 3450 m) in Switzerland. Erythropoiesis was evaluated in six randomly selected volunteers by performing peripheral blood mononuclear cell (PBMC) isolation at two time points: day 140 before altitude (Appendix A) and at day 19 of high altitude. Results from five volunteers were eventually employed for the final statistical analysis; the other volunteer was excluded due to low cell expansion yield, which impairs in vitro erythropoiesis.

### 2.2. PBMC Isolation and Culture

Peripheral blood (~25 mL) was collected in Li-heparin tubes (Sarstedt, Germany) for each culture condition: 20% O_2_ and 3% O_2_ for pre-altitude samples and 20% O_2_ for high altitude samples, named as 20% O_2_ JJ. PBMC were isolated using Ficoll Histopaque (density = 1.077 g/mL, 20 °C; GE Healthcare, Chicago, IL, USA) following the manufacturer’s protocol. Remaining RBC in the cell isolate were lysed (lysis buffer = 155 mM NH_4_Cl, 12 mM KHCO_3_, 0.1 mM EDTA; 10 min, room temperature). PBMC were cryopreserved at −80 °C in 5% dimethyl sulfoxide (DMSO, Sigma-Aldrich, St. Louis, MO, USA), 50% fetal calf serum (FCS, Biowest, Nuaillé, France) and 40% CellQuin medium (Sanquin, Amsterdam, The Netherlands) until use. Frozen PBMC were thawed, washed with phosphate-buffered saline (PBS) and cultured as previously described [18]. In short, a two-phase culture system was employed for in vitro erythropoiesis: in the expansion phase, PBMC (day 0 expansion) were cultured in CellQuin medium supplemented with EPO (2 IU/mL; Prospec, Ness Ziona, Israel), dexamethasone (1 µM; Sigma-Aldrich, St. Louis, MO, USA) and human stem cell factor (hSCF; 100 ng/mL, ITK diagnostics, Uithoorn, The Netherlands). The differentiation phase was started at day 13 expansion (day 0 differentiation) by washing the cells once with PBS and reseeding them in CellQuin supplemented with human plasma (5% *v*/*v*; Octapharma GmbH, Heidelberg, Germany), EPO (10 IU/mL), heparin (5 IU/mL; MP Biomedicals™, Irvine, CA, USA) and additional holotransferrin (final concentration = 1000 µg/mL; Sanquin, Amsterdam, The Netherlands). All cultures were kept in humidified incubators (37 °C, air + 5% carbon dioxide and 20% or 3% O_2_, according to the experiment). Culture media for cultures performed at 3% O_2_ were primed for a minimum of 1 h in the 3% O_2_ incubator to minimize the exposure of cells to atmospheric oxygen. Afterwards, culture dishes were removed from the incubator only for the necessary time to perform medium change and collections of samples for following tests. Cell concentrations were regularly determined using a CASY cell counter (CASY Model TCC; OLS OMNI Life Science, Bremen, Germany) together with cell volume measurements (Schärfe System GmbH, Reutlingen, Germany).

### 2.3. Flow Cytometry

Cultured cells were washed with PBS (5 min, 600× *g*), stained with primary antibody or reagents (20 min, room temperature), washed and resuspended in flow cytometry buffer (5 min, 600× *g*) and measured in a FACS Canto™II flow cytometer (BD Biosciences, Franklin Lakes, NJ, USA). Antibodies used were CD71-PB (1:100 dilution; Miltenyi Biotec, Bergisch Gladbach, Germany) and CD235a-PE (1:200 dilution; Biolegend, San Diego, CA, USA). Gating was done against specific isotype controls: anti-mouse isotype control IgG1k-PB (1:200; Biolegend, San Diego, CA, USA) and anti-mouse isotype control IgG1-PE (1:200;Thermo Fisher Scientific, Waltham, MA, USA). Enucleation was followed by nuclear staining with Deep Red Anthraquinone 5 (DRAQ5; 1:2500 dilution; cat#ab108410, Abcam, Cambridge, UK). For circulating hematopoietic stem cell detection, 10 µL of fresh blood was diluted in 1 mL PBS, stained with antibody anti-CD34 PerCP-Cy5.5 (BD Biosciences, Franklin Lakes, NJ, USA) for 25 min at room temperature and measured with a CyFlow Cube 5 N (5 million events per sample; Sysmex, Japan). The obtained data were analyzed with Flowjo™ (BD Biosciences, Franklin Lakes, NJ, USA).

### 2.4. Cytospin

About 100,000 cultured cells were washed with PBS and centrifuged onto glass slides using a cytocentrifuge (72× *g*, 5 min; Cytospin II Shandon, Marshall Scientific, Hampton, NH, USA). Cells were then stained for nucleus, cytoplasm and hemoglobin by difco red, difco blue (Sigma-Aldrich, St. Louis, MO, USA) and benzidine (1% o-dianisidine in methanol) [19].

### 2.5. Metabolites and Ion Measurements

For a general assessment of cell metabolism, supernatants of cell cultures were collected (600× *g*, 5 min), snap-frozen in liquid nitrogen and stored at −80 °C until measurement. Glucose, lactate, Na^+^, K^+^, Cl^−^ and Ca^2+^ were measured using a blood analyzer (Rapidlab 1265, Siemens, Munich, Germany).

### 2.6. Spectrophotometry on Hemoglobin

Hemoglobin content was determined as previously described [20]. Briefly, 150,000 cultured cells per sample were collected, pelleted (600× *g*, 5 min) in triplicate in a 96-well plate, and lysed in 20 µL distilled water. Prior to the measurement, 100 µL/mL of a reagent mix containing 0.5 mg/mL o-phenylenediamine dihydrochloride, 1 µL/mL hydrogen peroxide (H_2_O_2_) and 0.1 M citrate/phosphate buffer adjusted to pH = 5 was added to each well in a timed manner. After 2 min, the reaction was stopped by the addition of 20 µL/mL 8 M sulfuric acid (H_2_SO_4_) in each well, controlling so that the incubation time of each well with the reagent mix was the same. A full row in every plate was used for the addition of the reagents as a blank. The absorbance at 492 and 620 nm was measured using a BioTek Synergy 2 plate reader (Agilent, Santa Clara, CA, USA).

### 2.7. HPLC

At day 13 differentiation, a minimum of 5 million cells per sample were collected, washed in hydroxyethyl piperazineethanesulfonic acid (HEPES) buffer and resuspended in 1 mL PBS prior to the measurements. Hemoglobin isoform expression was determined by high-performance cation-exchange liquid chromatography (HPLC) on Waters Alliance 2690 equipment as previously described [21].

### 2.8. Deformability

Culture reticulocyte deformability (day 13) was measured by the Automated Rheoscope and Cell Analyzer (ARCA) as described previously [22]. Ten Pa shear stress was used, and 3000 cells were measured and grouped in 30 bins according to increasing elongation or cell projection area (as a measure of membrane surface area). Both the extent of elongation (major cell radius divided by minor cell radius) and the area (in mm^2^) were plotted against the normalized frequency of occurrence. Average deformability value and standard deviation were used to compare the three culture conditions of the five volunteers.

### 2.9. Cell Fixation and 3D Confocal Imaging

Samples at day 13 differentiation were collected, immediately fixed and imaged as previously described [23]. In short, 1 mL of 0.1% glutaraldehyde (Sigma-Aldrich, St. Louis, MO, USA) in PBS at room temperature was added to the cultured cells, which were preserved in this solution prior to imaging. After three washes in PBS for 4 min at 600× *g* to remove traces of glutaraldehyde, fluorescent cell staining was performed with 5 µL of CellMask Deep Red™ (0.5 mg/mL; Thermo Fisher Scientific, Waltham, MA, USA) in 1 mL of PBS for 24 h at room temperature. Following the other three washes, cells were resuspended in 1 mL PBS and placed between two glass slides together with 20 µm-diameter microbeads (Microbeads AS, Skedsmokorset, Norway) used as spacers. Confocal stacks of cells were obtained with a spinning disk-based confocal head (CSU-W1, Yokogawa Electric Corporation, Tokyo, Japan) for a 20 µm z-range with a 300 nm piezo-stepper. A solid-state laser emitting at 647 nm (Nikon LU-NV Laser Unit) was employed, and image sequences were acquired with a digital camera (Orca-Flash 4.0, Hamamatsu Photonics, Japan). Single cells were cropped and restacked with a custom MATLAB (MathWorks, Natick, MA, USA) program and transformed into isosurfaces for defined 3D cell reconstructions as previously described [24]. The 3D cells were visualized in Blender (Blender Foundation, Amsterdam, The Netherlands) for the manual count of the percentage of concave and biconcave cells over the total, going from a minimum of 400 to a maximum of 800 cells per sample.

### 2.10. Proteomics

#### 2.10.1. Sample Preparation and Mass Spectrometry (MS) Analysis

Samples containing 5 to 30 million cells were lysed in 100 µL of 4% sodium dodecyl sulfate (Sigma-Aldrich, St. Louis, MO, USA), 10 mM tris(2-Carboxyethyl)-phosphine HCl (Thermo Fisher Scientific, Waltham, MA, USA), 40 mM 2-chloroacetamide, 100 mM tris-HCl pH 8 (Invitrogen, Waltham, MA, USA) and by heating 5 min at 95 °C. After cooling to room temperature, 1U benzonase (Merck, Darmstadt, Germany) was added and the samples were sonicated for 10 min. Next, samples were centrifuged for 10 min at 10,000× *g*, and supernatants were transferred to fresh tubes. Protein content was determined with a Qubit protein assay (Invitrogen, Waltham, MA, USA), and 30 µg of protein was processed and digested as described previously [25]. Tryptic digests were acidified using 15 µL 2% formic acid (Thermo Fisher Scientific, USA) and desalted and fractionated in three fractions with StageTips containing three layers of styrenedivinylbenzene-reverse phase sulfonate (SDB-RPS) material (Empore^TM^, Sigma-Aldrich, St. Louis, MO, USA) as previously described [25]. Eluted peptides were vacuum dried and dissolved in 30 µL 0.1% trifluoroacetic acid (TFA; Thermo Fisher Scientific, Waltham, MA, USA), 2% acetonitrile (Biosolve, Dieuze, France). Three µL of peptides were separated by nanoscale C18 reverse chromatography coupled on-line to an Orbitrap Fusion Lumos Tribrid mass spectrometer (Thermo Fisher Scientific, Waltham, MA, USA) via a nanoelectrospray ion source at 2.15 kV. Buffer A was composed of 0.1% formic acid and buffer B of 0.1% formic acid and 80% acetonitrile. Peptides were loaded for 17 min at 300 nL/min at 5% buffer B, equilibrated for 5 min at 5% buffer B (17–22 min) and eluted by increasing buffer B from 5–27.5% (22–122 min) and 27.5–40% (122–132 min), followed by a 5 min wash to 95% and a 6 min regeneration to 5%. Survey scans of peptide precursors from 375 to 1500 *m*/*z* were performed at 120,000 resolution (at 200 *m*/*z*) with a 4 × 10^5^ ion count target. Tandem mass spectrometry was performed by isolation with the quadrupole, with isolation window 0.7, higher energy collisional dissociation (HCD) fragmentation with normalized collision energy of 30 and rapid scan mass spectrometry analysis in the ion trap. The tandem mass spectrometry (MS2) ion count target was set to 3 × 10^4^, and the max injection time was 20 ms. Only those precursors with charge state 2–7 were sampled for MS2. The dynamic exclusion duration was set to 30 s with a 10 ppm tolerance around the selected precursor and its isotopes. Monoisotopic precursor selection was turned on. The instrument was run in top speed mode with 3 s cycles. All data were acquired with Xcalibur software (Thermo Fisher Scientific, Waltham, MA, USA).

#### 2.10.2. MS Data Analysis

MS raw files were processed with MaxQuant 2.0.1.0 [26] using the human Uniprot database (downloaded March 2021). MaxQuant output tables were analyzed using R/Bioconductor (version 4.1.2/3.14) [27], “reverse”, “potential contaminants” and “only identified by site” peptides were filtered out and label-free quantification values were log2 transformed. Proteins quantified in 100% of an experimental group were selected for further analysis. Missing values were imputed by a normal distribution (width = 0.3, shift = 1.8), assuming these proteins were close to the detection limit. Statistical analyses were performed using moderated *t*-tests in the LIMMA package [28]. A Benjamini–Hochberg adjusted *p*-value < 0.05 and absolute log2 fold change > 1 was considered statistically significant and relevant. Raw MS files and search/identification files obtained with MaxQuant have been deposited in the ProteomeXchange Consortium [29] via the PRIDE partner repository with the dataset identifier PXD031776.

### 2.11. Statistical Analysis

Two-tailed paired *t*-tests performed in Prism 8 (GraphPad Software, San Diego, CA, USA) were used for comparison between atmospheric (20% O_2_) and low oxygen cultures (3% O_2_) or atmospheric oxygen from cells isolated before (20% O_2_) and from high altitude (20% O_2_ JJ). For non-normally distributed data, non-parametric two-tailed Mann–Whitney tests were performed. Plots show mean values and standard deviations from a total of five volunteers.

## 3. Results

### 3.1. In Vitro Erythropoiesis at 3% O_2_ Leads to a Reduced Erythroid Yield and Cell Volume Compared to That at 20% O_2_

We have previously shown the culture of erythroblasts and efficient differentiation to >90% enucleated reticulocytes from total PBMC [18]. We first investigated how O_2_ levels influence the in vitro culture process. Standard cultures are performed at 20% O_2_ at 37 °C, with dO_2_ = 7.2 mg/L roughly 12 times higher compared to in vivo bone marrow. Three percent O_2_ represents an oxygen saturation closer to the average of the extravascular space of the bone marrow (0.2–0.9 mg O_2_/L). The expansion phase was performed for 13 days, followed by a differentiation phase of another 13 days. Growth curves revealed a statistically significant lower cell yield at 3% O_2_ compared to 20% O_2_ throughout the entire culture time (Figure 1A). Cell volume was also decreased for cells at three percent O_2_ during the expansion phase, but no significant differences were found during differentiation (Figure 1B). The comparison between erythroid cultures from PBMC isolated from individuals pre-altitude (“20% O_2_” samples) and at high altitude (“JJ” samples) performed in standard oxygen conditions highlighted a higher erythroid yield in JJ samples (Figure 1C). Of note, a significant difference in CD34+ hematopoietic stem and progenitor cells (HSPC), the precursors of the erythroblasts in our culture system, between high altitude and post altitude measurements was observed for all the high-altitude JJ samples (Figure 1D). The increased number of mobilized HSPC within blood may explain the observed higher yield in the 20% O_2_ JJ samples [30]. The increase at day 1 for JJ samples and the decrease at day 1 post altitude suggest an immediate response on mobilization and homing of HSPC. No differences in volume changes were observed between 20% O_2_ JJ and 20% O_2_ (pre-altitude) and JJ (Figure 1E).

### 3.2. Erythroid Cultures at Three Percent O_2_ Show Accelerated Maturation

Flow cytometry performed during erythroblast differentiation was used to assess the differentiation state of the erythroid cultures at different oxygen pressures. Differentiation progression can be monitored by following the expression of the holo-transferrin receptor (CD71) and glycophorin A (CD235, MN blood groups). CD71^+^/CD235^−/low^ erythroblasts differentiate to CD71^+^ CD235^+^ basophilic erythroblasts, which gradually reduce the expression of CD71 to become CD71^-^ CD235^+^ ([31], Appendix A). No difference in the progression of CD71 and CD235 was observed both for the 3% vs. 20% and 20% vs. 20% O_2_ JJ cultures during differentiation (Figure 2A,B), except for CD71^−^ CD235^high^ at day 3 differentiation, which was higher for three percent O_2_ cultures (Figure 2A).

In contrast, a faster enucleation for cells at three percent O_2_ was observed during differentiation. Note that the total enucleation percentage at the end of differentiation (day 13) was similar between the cultures, suggesting that the erythroblast enucleation process itself was unaffected, but its dynamics were changed (Figure 2C–E). In fact, the ratio of reticulocytes over nuclei was higher for cells at three percent O_2_ (Figure 2E), suggesting an improved survival of reticulocytes formed at low oxygen or more unstable pyrenocytes. Enucleation in 20% O_2_ and 20% O_2_ JJ cultures was comparable throughout differentiation (Figure 2F–H). Together, the data suggest an increased state of differentiation of erythroid cultures at 3% O_2_ vs. 20% O_2_.

### 3.3. Hypoxia Accelerates Hemoglobinization during Cell Expansion with Increased HbF and HbA2 Expression in Differentiation

During in vitro erythroblast differentiation, cells massively upregulate globin expression to produce hemoglobin [32]. Total hemoglobin expression was followed during the first days of differentiation. A significantly increased hemoglobinization was found in cultures at 3% O_2_ compared to 20% O_2_ at the onset of differentiation (day 0), while this difference was not observed during the remainder of differentiation (Figure 3A). Although total hemoglobin levels are similar at the end of differentiation, HPLC data showed that cultures at three percent O_2_ had an increased expression of fetal hemoglobin (HbF) with a concomitant decreased level of adult hemoglobin (comprising HBA1 and HBB globin chains). In addition, the expression of HbA2, including HBD and HBA globin chains, was increased (Figure 3B and Appendix A).

The higher hemoglobinization at day 0 suggests that cells in cultures at three percent O_2_ were differentiating more during the expansion phase compared to erythroid cells cultured at 20% O_2_. This is in agreement with cytospins at day 0 showing the appearance of reticulocytes, which remained statistically different until day 6 (Figure 3C,D) but not at the end of differentiation, consistent with flow cytometry data (Figure 2C). As expected, terminally differentiated erythroid cells of 20% O_2_ JJ samples did not show such differences (Figure 3E), strengthening the observation that erythropoiesis immediately adapts to the higher oxygen levels ex vivo.

### 3.4. Deformability of In Vitro Reticulocytes Is Similar between 3% O_2_ or 20% O_2_ Cultures, but More Biconcave Cells Form at 3% O_2_

As a functional assay, the deformability of reticulocytes was assessed under controlled shear stress (Figure 4A). No significant differences were observed with respect to cell deformability in each culture condition. The obtained reticulocytes were additionally evaluated for morphological characteristics in 3D (Figure 4B). A significant difference was found between the cultures at different O_2_ percentages, with three times more concave and biconcave cells in cultures at 3% O_2_ compared to 20% cultures, suggesting a further maturation of reticulocytes towards red blood cells.

### 3.5. Metabolic Activity during In Vitro Erythropoiesis Is Changed between 3% and 20% O_2_ Conditions

Culture at various oxygen pressures may change the metabolic homeostasis of erythroid cells and balance between respiratory or anaerobic metabolism. Measurements of glucose and lactate concentration in cell culture medium were performed throughout differentiation (Figure 5). Glucose consumption was significantly higher for cells at 3% O_2_ compared to 20% O_2_ cultures (Figure 5A). In addition, significantly higher lactate production was found in three percent O_2_ erythroid cultures (Figure 5B). Higher lactate formation and increased glucose consumption at low oxygen concentrations may be a sign of increased glycolysis at the expense of respiratory, citric acid cycle metabolism during erythropoiesis. Assessment of ion content revealed a higher Na^+^/K^+^ ratio for cells at low oxygen (Figure 5C), suggesting a more elevated activity of the sodium–potassium pump, which contributes to maintaining cell membrane resting potential with consumption of ATP and has been described to be tightly linked to glycolytic activity [33]. Chloride content in the medium increased throughout differentiation for 20% O_2_ cultures, while it remained steady for cultures performed at three percent O_2_ (Figure 5D). There exist some differences between 20% O_2_ and 20% O_2_ JJ cultures at day 0 for glucose, Na^+^/K^+^ and chloride, which are reduced in the following days until no difference is found by day 13.

### 3.6. Proteomics on In Vitro Reticulocytes Reveals Metabolic Adjustments at Different O_2_ Concentrations

Erythroid cultures at different O_2_ pressures may cause specific changes to the proteome of the terminally differentiated reticulocytes. Proteomics analysis was performed on cultures at day 13 differentiation on four volunteers in the three culture conditions. Between 1200 and 1800 distinct proteins were quantified in the reticulocytes (Figure 6A; Appendix A). Dimensional reduction using the first two principal components (PC) explain ~65% of the variation between the different sample cohorts. PC2 described the volunteer–volunteer variation, while PC1 (42%) described the variation due to the different culture conditions between the three cohorts (Figure 6B). While 20% O_2_ vs. 20% O_2_ JJ samples were highly similar (Figure 6C), 20% O_2_ samples and 3% O_2_ showed differences. Twenty, 190 and 106 differentially abundant proteins were observed between 3–20% O_2_; 3–20% O_2_ JJ and 3–20% O_2_ total (in which all 20% cultures were pooled together), respectively (Figure 6C; Appendix A). A heatmap with all significant proteins in each group was generated, and three clusters were identified (Figure 6D). The majority of the significantly different proteins were decreased in abundance in the 3% O_2_ culture condition compared to 20% O_2_. Interestingly, the 20% O_2_ JJ samples displayed significantly more differentially abundant proteins with the 3% O_2_ than the 20% O_2_ control cultures. Such proteins function within mitochondrial metabolic pathways, ribosomal biogenesis and cell cycle processes as indicated by KEGG and STRING analysis (Appendix A). Indeed, 6 out of the 11 glycolytic enzymes were more abundant in three percent O_2_ erythroid cultures, while the remaining enzymes displayed a similar trend (Figure 7A). This increased glycolytic enzyme expression, among which LDHA may provide a possible explanation for the increased lactate production observed in Figure 4. Concomitant with increased enrichment of glycolytic enzymes in three percent O_2_ culture conditions, citric acid cycle enzymes decreased, of which PDHA1, IDH2, OGDH and CS were statistically significant. Notably, PDHA1 is the enzyme catalyzing the production of acetyl-CoA from pyruvate, crucial to kick start the citric acid cycle (Figure 7B). Of note, the pentose phosphate pathway to regenerate NADPH, protecting against oxidative stress in reticulocytes and RBCs, was unchanged between 20% and 3% O_2_ cultures (Figure 7C). Similarly, enzymes in the glutathione cycle were unchanged (Appendix A). The data indicate that enzymes involved in glycolysis and the citric acid cycle are differentially enriched upon culture in different O_2_ concentrations in a reciprocal manner.

### 3.7. Proteins Involved in the Structural Integrity of Erythrocytes Are Unchanged between Cultures at Different O_2_ Percentages

We found that reticulocytes cultured at different O_2_ pressure do not display changes in deformability (Figure 4A). This suggests that proteins involved in the structural integrity and flexibility of erythrocytes are possibly unaffected (for review see [34]). The proteomic dataset allowed us to analyze the expression of the main structural proteins. As expected, proteins comprising the junctional and ankyrin complex, ankyrin (ANK-1), spectrin (SPTA1, SPTB), protein band 3 (SLC4A1), protein band 4.1 (EPB41) and 4.2 (EPB42), tropomyosin (TPM1), adducin (ADD2), GLUT-1 (SLC2A1), RhAG, P55, stomatin and dematin (DMTN) were unchanged (Figure 8A). In addition to these structural proteins, CD71 (TFRC) expression was lowered at three percent O_2_, but not statistically significant (Figure 8A). Expression of CD55 and CD59 was not detected, and CD47 was identified but not quantifiable in this dataset. In agreement with the HPLC data, gamma globin subunits 1 (HBG1) and 2 (HBG2), partly comprising fetal hemoglobin, were more abundantin three percent O_2_, with HBG1 significantly increased. In addition, hemoglobin subunit mu (HBM) was found to be increased, while hemoglobin beta (HBB), alpha 1 (HBA1) and delta (HBD) remained unchanged (Figure 8B). These data suggest that cultures at three percent O_2_ do not impact erythrocyte structural stability but do affect metabolic enzyme and globin subunit expression.

## 4. Discussion

This study aimed at understanding the impact of O_2_ levels on erythropoiesis to elucidate both the physiological adaptations of hematopoietic cells to high altitude and the experimental implications of oxygen concentration on in vitro erythropoiesis. In vitro erythropoiesis was performed at atmospheric oxygen (20% O_2_) and at low oxygen (3% O_2_), where the latter is comparable to the physiological concentrations in the bone marrow [1]. Interestingly, we observed a direct response of erythroblasts to the available oxygen at different stages of maturation. At an elevated concentration (20% O_2_ in both pre-altitude and 20% O_2_ JJ samples), erythroblasts expanded faster and displayed increased cell volume compared to cells cultured at three percent O_2_. Erythroblast cultures at the onset of differentiation showed signs of accelerated differentiation at three percent O_2_, including increased hemoglobin expression and enucleation percentage (cytospins, flow cytometry) as well as premature CD71 loss. Importantly, however, these differences were not observed in the final, terminally differentiated cultured RBC. The ratio of reticulocytes over nuclei was lower in 20% O_2_ cultures, which we suggest may be either due to improved reticulocytes survival at three percent O_2_ or a higher instability of the expelled nuclei (pyrenocytes) at low oxygen. A more than doubled amount of concave shapes in cells grown at low oxygen was observed. Reticulocytes initially form as irregular shapes with several lobes [35] that during maturation undergo morphological changes until the formation of a biconcave disk [36]. During reticulocyte maturation a spectrum of different morphological configurations occurs. These processes are as yet ill-defined mechanistically but involve restructuring of membrane protein complexes and their dynamic interaction with the underlying spectrin cytoskeleton, as well as regulation by post-translational protein modifications, loss of specific proteins through vesicle release (e.g., AQP1 and CD71) and shear stress signaling [36,37,38,39]. The presence of proto-biconcave cells, meaning a large biconcave cell with irregular borders, suggests a more advanced maturation stage of the reticulocytes formed at three percent O_2_. In agreement with this, CD71 expression was slightly decreased in 3% O_2_ cultures compared to 20% cultures, but not a statistically significant amount. Importantly, results on reticulocyte deformability showed no functional differences among all culture conditions, and indeed, similar expression of the main cytoskeletal proteins was observed.

Interestingly, a slight increase in erythroid yield was observed in 20% O_2_ JJ samples compared to control 20% O_2_ cultures. Within the PBMC isolated from the JJ samples, we observed a significant increase in the percentage of mobilized HSPC in blood. This increased HSPC in PBMC, the starting point of the in vitro cultures [18], possibly explains this observed increased erythroid yield. The origin of the increased HSPC mobilization is currently unknown but has been reported before [40] and is not exclusive to HPSC but is also observed for neutrophils, other myeloid cells [41] and burst-forming unit-erythroid (BFU-e) precursors concomitant with increases in plasma EPO levels [42]. It may be caused by a higher blood flow in the bone marrow occurring during hypoxia that induces stem cells’ entrance in the circulation. In addition, HSPC mobilization is tightly controlled by an interplay between cell–cell interactions and chemokines (e.g., SDF1) and also regulated by atypical chemokine receptor (DARC, containing the duffy antigen) expression on erythroid cells [43]. We have recently shown that DARC expressed on nucleated erythroblast binds SDF1 but that this is lost upon reticulocyte maturation [39]. Increased erythroid mass due to hypoxia-induced stress erythropoiesis may signal mobilization of HSPC. It will be interesting to assess the plasma levels for specific mobilizing chemokines upon systemic hypoxia. Nevertheless, the results suggest that from a morphological, deformability and structural point of view, erythropoiesis at low oxygen and from high altitude HSPC is similar to controls cultured at 20% O_2_. This suggests that circulating hematopoietic stem and progenitor cells isolated at high altitude were either unaffected by hypoxia or that any possible changes (e.g., epigenetic modifications) are quickly modified upon culture at 20% O_2_.

We observed an increase in hemoglobin expression during the expansion phase in cells at low oxygen. Interestingly, HbF and HbA2 expression as well as the expression of the globin subunit HBG1 (proteomics) was increased in enucleated reticulocytes at three percent O_2_. It has been described that HbF and HbA2 are relatively more expressed in immature erythroblasts compared to later differentiation stages where HbA is predominantly expressed [44,45]. As overall hemoglobinization is unchanged at the end of terminal differentiation, the increased expression of HbF and HbA2 may be a sign of differential early transcriptional activation of these globin genes during erythroblast expansion and differentiation at low oxygen tension. Although the present study does not identify mechanisms behind this regulation, the data does suggest a link between O_2_ concentration and specific globin regulation. Of note, increased HbF expression may also be a physiological response to hypoxia. HbF has a higher oxygen affinity than HbA, contributing to improved oxygen delivery to the peripheral tissues, where the higher acidity caused by the enhanced tissue hypoxia may favor HbF oxygen release due to the Bohr effect [46,47,48].

The main changes between cultures at low and high oxygen were related to cell metabolism and ionic equilibriums. RBCs are dependent on glycolysis for ATP production due to autophagy of organelles (including mitochondria) during terminal differentiation, necessitating a metabolic switch from oxidative respiration to glycolysis. Indeed, extracellular lactate production increased and glucose decreased during differentiation in both 20% and 3% O_2_ conditions, confirming this switch. However, glucose consumption was significantly higher in 3% O_2_ cultures compared to 20% O_2_ and was accompanied by higher lactate levels. Proteomics data of terminally differentiated reticulocytes confirmed upregulation of glycolytic enzymes at three percent O_2_ cultures and revealed downregulation of the enzymes involved in oxidative respiration. These results suggest that changes in metabolic pathways regulate cellular processes such as cell division and cell volume [49,50,51]. These adaptations to hypoxia are linked to the activity of HIF factors, of which HIF-1-alpha is known to promote HSC quiescence and increase glycolytic enzyme expression [49,52].

The higher anaerobic production of ATP occurring in cells at three percent O_2_ may explain the higher ratio of Na^+^/K^+^ in the culture medium as a result of greater activity of the Na^+^/K^+^ pump, which has been linked to higher glycolytic activity [33]. This pump increases its activity according to the rate of ATP formation [33]. Hypoxia is a trigger for ATP release from RBCs, which is thought to stimulate NO production by endothelial cells [53,54]. The mechanisms for ATP release in RBCs are not fully understood, but must involve channels with large pores, between 0.6 and 1.1. nm, which allow permeation of small ions, such as Na^+^ and K^+^ [55]. Thus, increased pump activity to restore membrane resting gradient would be required [56]. Of note, the increase in its activity was not associated with an increase in the expression of the two subunits comprising the ATP-dependent Na^+^/K^+^ pump, ATP1B3 and ATP1A1 (Appendix A).

Lactate formation as a consequence of glucose metabolism through glycolysis results in cell culture medium acidification and possibly leads to increased activity of the Na^+^/H^+^ exchanger to remove H^+^ and introduce Na^+^ in the cytoplasm, further justifying a need to increase Na^+^/K^+^ pump activity. This exchanger is coupled with the anion exchanger band 3, which extrudes carbonate ions HCO_3_^−^ with Cl^−^ entering the cell. This could explain the lower Cl^−^ content in the medium found in the cultures performed at low oxygen. In addition, acidification causes cell swelling by osmotic effects, due to the inability of the Na^+^/H^+^ exchanger to regulate the formation of H^+^. Notably, proteomics investigations did not show differences in the levels of any of such transporters (Appendix A). Reticulocytes at atmospheric oxygen rather direct glucose consumption for reduction of thiols, the main protection against oxidative damage in RBC and reticulocytes [57]. In RBC, NADPH is exclusively generated via the pentose-phosphate pathway (PPP). The PPP is in competition with glycolysis and is regulated by the oxygenation state of hemoglobin [57]. Glycolytic enzyme upregulation was detected for cultures at low oxygen, but enzymes involved in the PPP remained similar, suggesting that regulation of NADPH production was untouched. However, as the PPP pathway is in direct competition with glycolysis, the upregulation of the latter may suggest a lowered PPP shunt in cells at low oxygen and an adaptation of increased PPP in 20% O_2_ cultures, where protection against reactive oxygen species would be more relevant.

The data showed that in vitro erythropoiesis at 3% or 20% O_2_ yielded reticulocytes with comparable viability and deformability, two aspects reported to be affected in RBCs formed during systemic hypoxia and stress erythropoiesis and implicated in inducing neocytolysis. As such, the data obtained in this study do not support the neocytolysis hypothesis, at least in terms of reticulocyte development and stability. Moreover, erythroblast differentiation and the resulting cultured red blood cells at low oxygen tension were highly similar to 20% O_2_ culture conditions. On the one hand, sample collection and culture media changes were performed quickly at atmospheric oxygen, causing temporary increases in the dissolved oxygen levels in the culture medium. A full control of oxygen tension (in bioreactor setups or in a controlled O_2_ workstation) may result in more pronounced differences in erythroid maturation between 3% and 20% O_2_ cultures. On the other hand, cell sampling during differentiation was performed every three days. Once an equilibrium with the oxygen tension of the incubator is established, the dissolved oxygen concentration in non-stirred static culture dishes may not be directly similar to the oxygen pressure within the incubator, as this depends on diffusion. The dissolved oxygen at the bottom of the dish may thus be considerably lower than the three percent O_2_ within the incubator [58]. However, this further fortifies the result that lower oxygen pressures support in vitro erythropoiesis. Of note, decreased expression of CD55, CD59 and CD47 has been associated with RBC senescence and was previously found to be downregulated upon decreasing EPO plasma levels after descent from high altitude [8,9]. Any role of oxygen influencing the expression of these markers could not be established, as CD55 and CD59 were not detected by proteomics, while CD47 was identified but not quantifiable.

In conclusion, our data are in agreement with our previous high altitude study confirming the absence of neocytolysis, which was verified by comparing the clearance time of age-cohort labeled RBC before altitude (110 m) and during a 19-day stay at high altitude (3450 m) and resulted in no differences. In that study, we demonstrated that the fast decrease in red blood cell mass upon return to sea level is solely caused by a reduced rate of erythropoiesis, while red blood cell clearance rate remains stable [11]. This is in accordance with the data presented here, in which in vitro characterization of erythroid maturation did not identify differences that could suggest a faster removal of RBCmatured at lower oxygen compared to atmospheric oxygen.

A final consideration should be addressed regarding in vitro erythropoiesis. The dissolved O_2_ concentration must be experimentally determined during upscaling to large processes (e.g., bioreactors). Importantly, the results in this study indicate that both the expansion as well as the differentiation at low oxygen tension is possible, and that resulting cultured RBC are highly similar. Lower O_2_ concentration during differentiation might contribute to in vitro RBC development beyond the reticulocyte state into mature RBC and allow more cost-effective cultures by reducing O_2_ consumption and sparging.

## 5. Conclusions

The data obtained show that in vitro erythropoiesis at three percent O_2_ resulted in a successful terminal differentiation to reticulocytes. Features of cultures performed at lower oxygen are a reduced expansion and cell volume, accelerated differentiation, retention of HbF expression, development of more concave reticulocytes and increased glycolytic activity. No differences were found in terms of cell deformability or levels of cytoskeletal proteins and enzymes involved in the protection from oxidative stress. Moreover, no difference was detected in cultures from stem cells isolated at high altitude (20% O_2_ JJ), excluding any retention of a hypoxic fingerprint. Overall, the data on in vitro erythropoiesis do not support the neocytolysis hypothesis.

## Figures and Tables

**Figure 1 cells-11-01082-f001:**
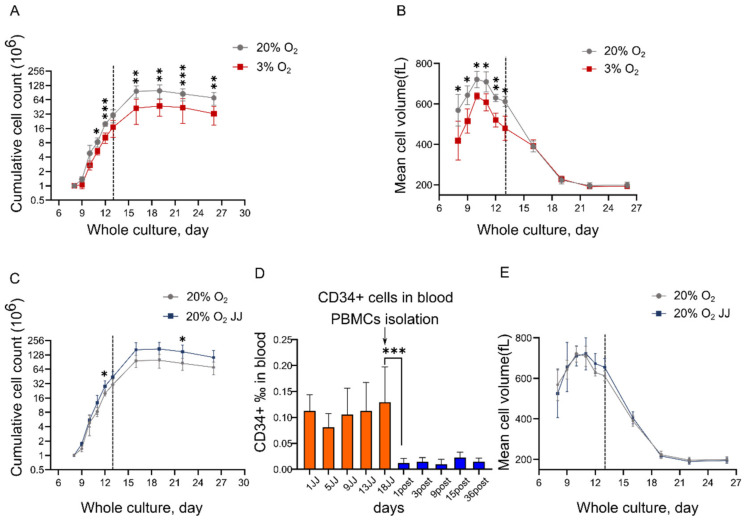
In vitro erythropoiesis performed at 3% O_2_ results in a lower erythroid yield and cell volume compared to cultures at 20% O_2_. (**A**) Cumulative growth curve of erythroid cultures from PBMC to enucleated reticulocytes (day 26). Grey dashed line indicates the change from the expansion phase to the differentiation culture condition (day 13 = day 0 differentiation), as outlined in Materials and Methods. (**B**) Mean cell volume throughout erythroid cultures was significantly lower in cells at 3% O_2_ during expansion but not during differentiation. (**C**) High altitude (20% O_2_ JJ) cultures resulted in an increased erythroid yield compared to pre-altitude (20% O_2_) samples. (**D**) Analysis of CD34+ cells revealed a higher percentage of HSPC circulating in peripheral blood at high altitude. These measurements were performed on all 12 volunteers involved in the high-altitude study. (**E**) Mean cell volume between 20% O_2_ and 20% O_2_ JJ samples was comparable. Data on cultures represent results of 5 of the volunteers involved in the high-altitude study (Materials and Methods). Statistical significance was determined using unpaired student *t*-test, with * = *p* < 0.05, ** = *p* < 0.01 and *** = *p* < 0.001.

**Figure 2 cells-11-01082-f002:**
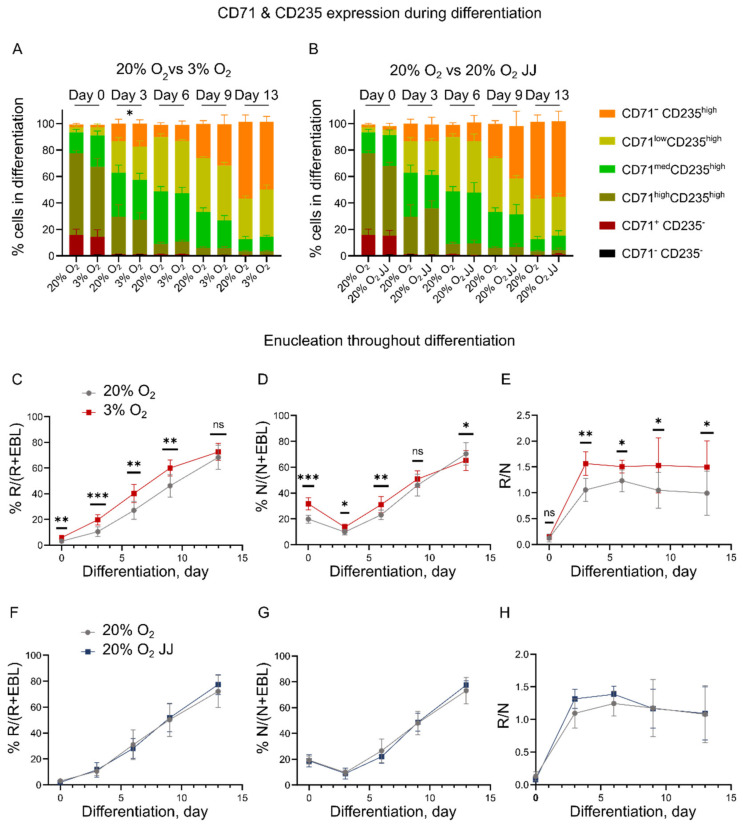
Erythroid marker expression and enucleation during differentiation. (**A**,**B**) CD71 and CD235 expression did not differ across all culture conditions, showing comparable populations. An exception is CD71–CD235^high^ cells, which were significantly higher at day 3 differentiation (*p* = 0.05) in 3% O_2_ cultures, but not on the following days, suggesting a faster differentiation that did not result in a higher number of formed reticulocytes by day 13. (**C**–**E**) Enucleation efficiency was evaluated by comparing the percentage of reticulocytes over the total cell number (% R/(R+EBL)) in (**C**), nucleated cells over the total cell number (% N/(N+EBL)) in (**D**) and the ratio of reticulocytes over nuclei (R/N) in (**E**). These data highlight a higher enucleation at 20% O_2_, but a better survival of the formed reticulocytes or unstable pyrenocytes at low oxygen. (**F**–**H**) Similarly, comparative graphs between 20% O_2_ and 20% O_2_ JJ cultures showed no difference either in enucleation and reticulocyte survival. ns=non-significant, * = *p* < 0.05, ** = *p* < 0.01 and *** = *p* < 0.001.

**Figure 3 cells-11-01082-f003:**
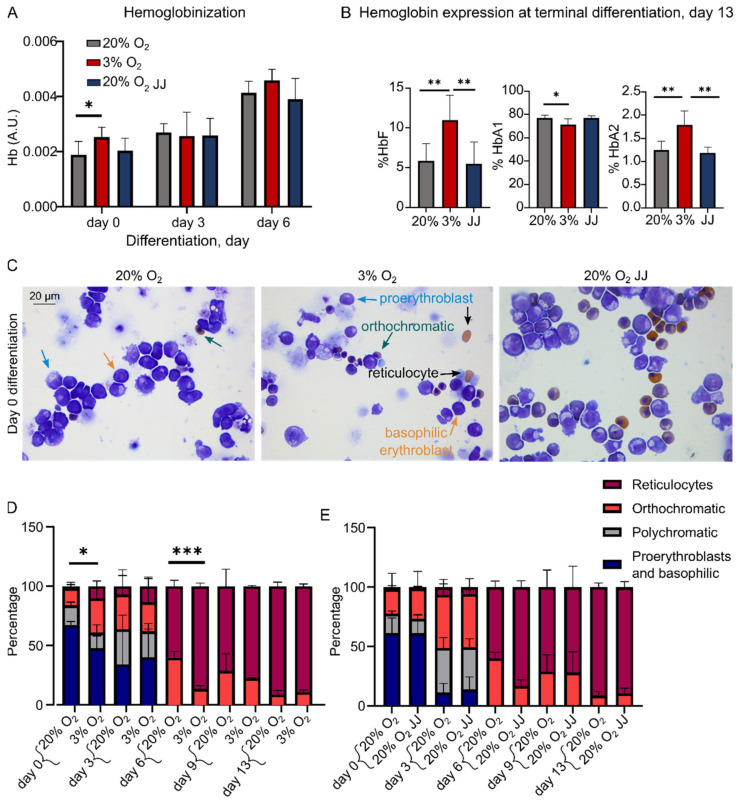
Hemoglobin expression is increased at onset of erythroblast differentiation with a changed globin composition. (**A**) Quantification of Hb levels determined by spectrophotometry (Materials and Methods) showed a significant increase at day 0 for cells at 3% O_2_. (**B**) HPLC analysis of hemoglobins at the end of differentiation (day 13). HbF expression was doubled in cells at 3% O_2_, reflected by a lower expression of HbA1. Moreover, HbA2 expression was retained in cultures performed at 3% O_2_. (**C**) Cytospins showed formed reticulocytes at 3% O_2_ at day 0, when cells were still suspended in the expansion medium. Occasional orthochromatic erythroblasts were found in 20% O_2_ and 20% O_2_ JJ cultures. (**D**) Statistical analysis on such images revealed a higher number of formed reticulocytes at day 0 and day 6 differentiation on a count of about 300 cells per volunteer. (**E**) No differences were found between 20% O_2_ and 20% O_2_ JJ. * = *p* < 0.05, ** = *p* < 0.01 and *** = *p* < 0.001.

**Figure 4 cells-11-01082-f004:**
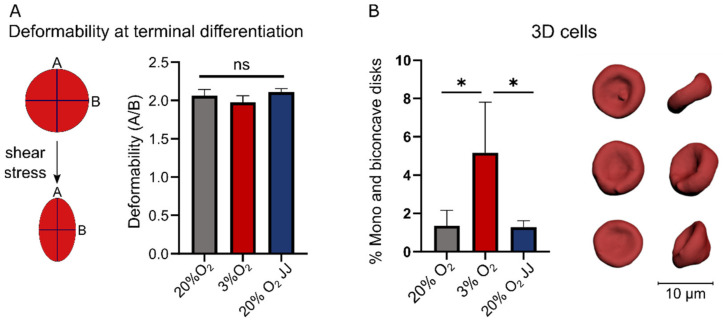
Deformability and 3D morphology of formed reticulocytes at terminal differentiation. (**A**) A functional assay measuring the deformability of reticulocytes revealed no differences between all culture conditions. (**B**) 3D confocal imaging highlighted on average three times more reticulocytes with a concave or biconcave shape in 3% O_2_ cultures. Front and side views of three representative reticulocytes from one volunteer are given. ns = non-significant, * = *p* < 0.05.

**Figure 5 cells-11-01082-f005:**
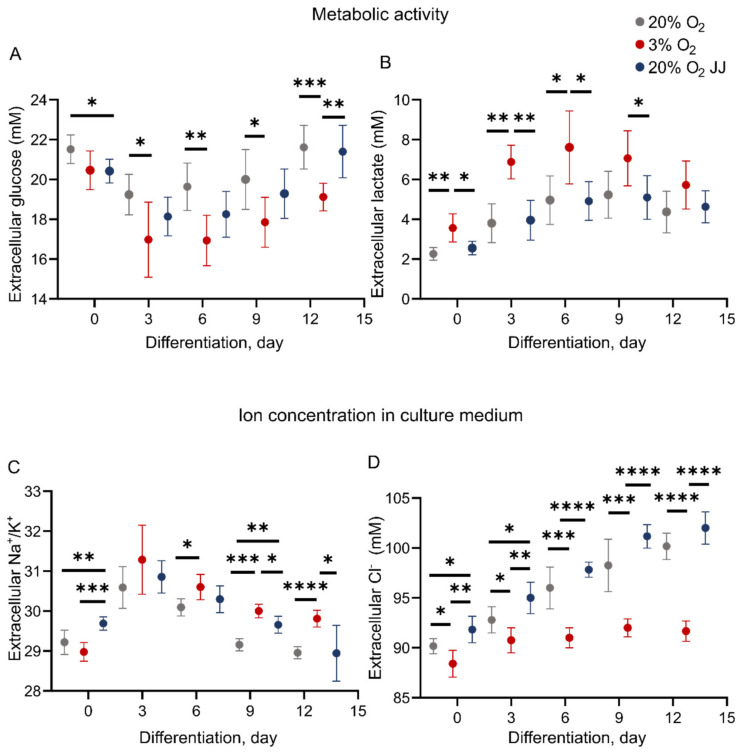
Metabolic activity and cell medium ion content during erythroblast culture. (**A**) Glucose consumption is more elevated for cells at 3% O_2_, suggesting a switch to glycolytic pathway and resulting in a higher production of lactate (**B**). (**C**) The ratio Na^+^/K^+^ in the cell medium was higher, with two significant time points, in cells at 3% O_2_. (**D**) Cl^−^ content, while remaining stable during differentiation of cells at 3% O_2_, resulted in daily increases for cells cultured at 20% O_2_. * = *p* < 0.05, ** = *p* < 0.01 and *** = *p* < 0.001.

**Figure 6 cells-11-01082-f006:**
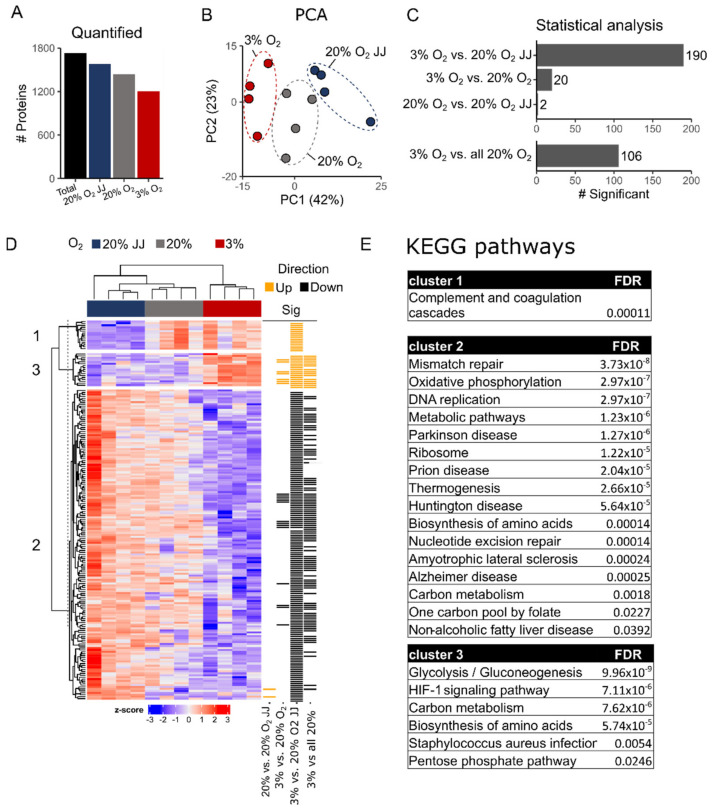
Proteomics analysis. (**A**) Amount of quantified proteins per experimental group and in the total proteomics dataset. (**B**) Principal component analysis (PCA) of 20% O_2_ JJ, 20% O_2_ and 3% O_2_ cultured erythrocyte proteome data. (**C**) Bar plot of number of statistically significant proteins. Comparison between all conditions (upper panel). Due to the limited differences between 20% O_2_ and 20% O_2_ JJ an additional analysis comparing all 20% O_2_ cultured cells with 3% O_2_ was performed (lower panel). (**D**) Heatmap of all statistically differentially abundant proteins. Z-scored label-free quantification intensities are shown as a gradient (low: blue, high: red). Row annotations indicate in which comparison a protein was significant (yellow: up, black: down). Numbers indicate cluster assignment. (**E**) KEGG analysis shows the pathways involved in each cluster, mostly highlighting metabolic changes in mitochondrial activity, ATP formation, glycolysis and gluconeogenesis. For further details on the connection between proteins see Appendix A.

**Figure 7 cells-11-01082-f007:**
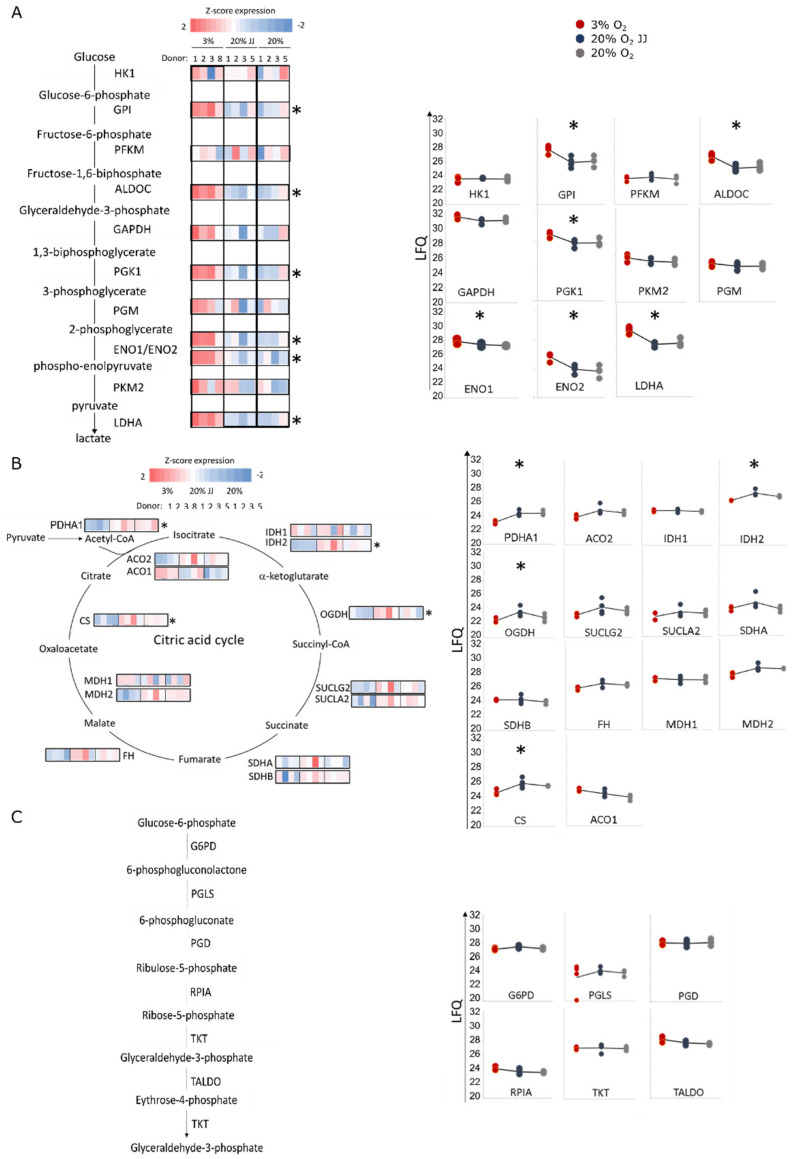
Proteomics data on metabolic pathways. (**A**) Glycolysis pathway, Z-scored LFQ values of each glycolytic enzyme show that six out of 11 enzymes are significantly more abundant at low oxygen, overall indicating that glycolysis is upregulated at 3% O_2_. (**B**) On the contrary, citric acid cycle enzymes are decreased, with four significantly different enzymes, pointing out a downregulation at low oxygen. (**C**) By contrast, pentose-phosphate pathway does not show any difference in any culture conditions. * = *p* < 0.05.

**Figure 8 cells-11-01082-f008:**
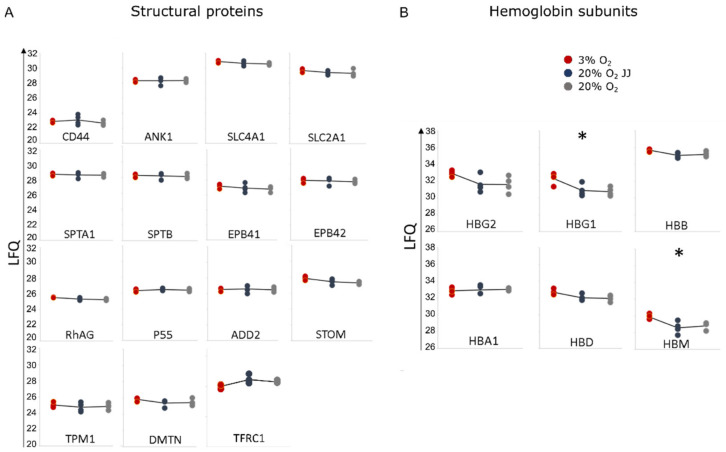
Proteomics data. (**A**) Typical structural proteins for reticulocytes and RBC stability. No significant differences were detected among the three culture conditions. (**B**) Hemoglobin subunits confirm a significantly higher expression of the gamma globin 1 (HBG1) and show a higher expression of hemoglobin subunit mu (HBM). * = *p* < 0.05.

## Data Availability

The data presented in this manuscript are available in the main Figures, Materials and Methods section and in the Appendix A. Raw MS files and search/identification files obtained with MaxQuant have been deposited in the ProteomeXchange Consortium via the PRIDE partner repository with the dataset identifier PXD031776. Additional measurements can be obtained upon reasonable request to the corresponding authors.

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
