# Peer review of "In Vitro Erythropoiesis at Different pO2 Induces Adaptations That Are Independent of Prior Systemic Exposure to Hypoxia"

_cells, 2022, doi:10.3390/cells11071082_

Round 1

Reviewer 1 Report

This is a very interesting well designed and expertly conducted study that assess the effect of  low oxygen on in vitro production of red blood cells.\

As stated by the authors ,the effect of low oxygen concentration and erythropoiesis is not well understood, and this study will help fill this gap.

I have no major comments or concerns because the manuscript is really clearly written and because the conclusions are very well supported by the data.

I only have a few minor comments.

It is a pity that study was not conducted in a O2 workstation  (biospherix, for instance) as it is well known that cells takes a very long time to re-adapt to low oxygen when they are move between atmospheric and hypoxia. More effects of hypoxia might have been observed if the cells had been kept i in hypoxia at all time (as happens in vivo)

It would be interesting to know  if there are more mature RBCs (i.e cells with no RNA) in 3% O2  as compared to atmospheric O2, as the loss of RNA is with the acquisition of a concave shape the hallmark of reticulocyte maturation.

It would have been interesting to present some data by flow or some other methods related to expression of the epoR and of CD117.

a figure depicting the results for the HPLC  hemoglobin quantification would be useful.

Discussion is fine. The data presented do not support the neocytolysis hypothesis. However, given the long intro on neocytolysis and the controversy, a couple of sentences stating what is the view of the authors related to the speed of return to normal  hematological value after space flight or coming back to sea levels after a stay in altitude would make the paper more interesting. Do the authors think that that there is anything to explain at all or not? Are the usual mechanism to remove aging red cells sufficient to explain the decrease in red cell mass and hematocrit after a stay or altitude ? or do they have some other hypothesis ? 

There are a few sentences that are not clear:

line 282: "may lay fundament", not standard english could be replace by "explain"

Line 345: sentence is not clear.

In the text figure 5 is mislabeled as figure 4

Some of the abbreviation are not defined in the methods

Author Response

Dear reviewer 1, please find our responses in the attachment.

Reviewer 2 Report

This is a very well conducted study of human erythropoiesis that compares the effects of low oxygen and high altitude. The numbers are good - 12 healthy young donors that travelled to high altitude. The methods are very well described and very well undertaken by an expert team. The results are presented clearly with strong statistics. I have a few specific questions that may or may not be able to be addressed in discussion of the results or overall discussion.

In Figure 1, why is the MCV lower in the expansion phase in low oxygen? Is there a hypothesis about mechanism?

Why are there more CD34+ cells in the blood at high altitude. I guess this is covered to some degree in the discussion. Are there also more dedicated BFU-e or CFU-e in the blood also? If there is any literature on this it should be discussed. 

Figs 2 and 3 are clear. 

For Fig 4, I am not sure of the significance of the increased biconcave erythrocytes in the low oxygen environment. I think this just reflects faster maturation but a clear statement about this would be helpful.

All of the proteomics has been done very well

I think the lack of observation of a neocytolysis effect is a strength, not a weakness. The data supporting this concept are not strong, so this paper will refute the concept. 

In short, this is a very well performed study and would be of interest to the readers of Cells

Author Response

Dear reviewer 2, please find our responses in the attachment.
